# Adaptive Stimulations in a Biophysical Network Model of Parkinson’s Disease

**DOI:** 10.3390/ijms24065555

**Published:** 2023-03-14

**Authors:** Thomas Stojsavljevic, Yixin Guo, Dominick Macaluso

**Affiliations:** 1Department of Math and Computer Science, Beloit College, 700 College St., Beloit, WI 53511, USA; 2Department of Mathematics, Drexel University, Philadelphia, PA 19104, USA; 3Department of Neurosurgery, University of Pennsylvania Health System, Philadelphia, PA 19104, USA

**Keywords:** Parkinson’s disease, deep brain stimulation, adaptive stimulation, multi-site stimulation, thalamocortical relay, basal ganglia model, local field potential

## Abstract

Deep brain stimulation (DBS)—through a surgically implanted electrode to the subthalamic nucleus (STN)—has become a widely used therapeutic option for the treatment of Parkinson’s disease and other neurological disorders. The standard conventional high-frequency stimulation (HF) that is currently used has several drawbacks. To overcome the limitations of HF, researchers have been developing closed-loop and demand-controlled, adaptive stimulation protocols wherein the amount of current that is delivered is turned on and off in real-time in accordance with a biophysical signal. Computational modeling of DBS in neural network models is an increasingly important tool in the development of new protocols that aid researchers in animal and clinical studies. In this computational study, we seek to implement a novel technique of DBS where we stimulate the STN in an adaptive fashion using the interspike time of the neurons to control stimulation. Our results show that our protocol eliminates bursts in the synchronized bursting neuronal activity of the STN, which is hypothesized to cause the failure of thalamocortical neurons (TC) to respond properly to excitatory cortical inputs. Further, we are able to significantly decrease the TC relay errors, representing potential therapeutics for Parkinson’s disease.

## 1. Introduction

Deep brain stimulation (DBS) is a therapeutic option for Parkinson’s disease (PD) wherein a stimulating electrode is surgically implanted into the subthalamic nucleus (STN), globus pallidus (GP), or thalamus (TC), depending on the more prominent symptoms. The stimulating electrode sends electrical impulses to the targeted regions of the brain [1,2,3]. At present, the most conventional DBS is done with open-loop methods involving constant high-frequency stimulation (HF). On a neuronal level, the pathophysiology of PD is associated with changes such as increases in synchrony, firing rates, and bursting activity in the basal ganglia [3,4,5,6]. DBS may achieve its therapeutic benefits when stimulation is able to disrupt the pathological synchronized bursting in the basal ganglia [7,8,9,10]. While this treatment has been effective, there are several drawbacks to HF DBS. In open-loop methods, the stimulation parameters such as duration, amplitude, and frequency of the pulse train are controlled by external forces that are not guided by the changes in the brain’s electrical activity resulting from PD [11]. Conventional HF DBS, with high frequency and nonadjustable parameters, may have adverse effects in the proximate region to the stimulation site. Additionally, battery and device shelf-life concerns are present with higher frequency and open-loop DBS [11,12].

A possible remedy is to develop a closed-loop, adaptive system that only administers DBS as necessary. A closed-loop method is a system in which DBS is automatically controlled in accordance with a recorded feedback signal from a targeted region. In this approach, stimulation is administered only when necessary and is, to an extent, dependent on the measured neuronal activity [13]. Due to the desire for a closed-loop, adaptive system, in vitro and in silico studies have been conducted. In 2013, Little et al. used brain–computer interfaces to monitor the local field potential (LFP) of the STN and to control the administration of DBS to patients with PD. Their results showed statistically significant improvement in motor scores [14]. Later, de Castro et al. employed delayed feedback control algorithms to disrupt unwanted pathological neuronal oscillations within in vitro networks [15]. Current experimental studies aim to identify appropriate biological feedback signals, such as neural oscillating or using the magnitude of the LFP [16,17,18,19,20].

Since other biological signals may be used to monitor and control the administration of adaptive deep brain stimulation (aDBS), we turn to computational models to explore potential avenues. Previous closed-loop computational modeling efforts at DBS have focused on using the amplitude of LFP. In particular, Guo and Rubin studied a multi-site delay feedback DBS in 2011 based on the LFP of the STN population [5,21,22,23]. Their multi-site delayed feedback stimulation was found to be more beneficial compared to open-loop HF. However, this method lacks an adaptive mechanism because the stimulation remains on the entire time once it is turned on. The primary recent computational studies in aDBS have focused on desynchronizing the STN neurons in parkinsonian networks [13,24,25,26,27,28,29,30,31,32,33,34,35,36]. Based on work conducted by Popovych and Tass in 2019, they concluded that using the LFP as a signal to control aDBS can be more efficient in suppressing abnormal synchronization than continuous stimulation [13]. Our work aims to look beyond the desynchronization of PD networks using aDBS. Here, PD networks are computational models of neurons that mimic neuronal patterns of PD in the basal ganglia.

We introduce two novel closed-loop aDBS techniques utilizing the interspike time of model STN neurons to gauge PD neuronal activity and to determine when stimulation is on and off. This will be applied to a small network of conductance-based STN, GP, and TC neurons that, by design, generate parkinsonian activity patterns in the absence of stimulation. While other authors have explored how adaptive stimulation techniques desynchronize neuronal networks in studying PD, we find that our closed-loop aDBS is able to both desynchronize and deburst parkinsonian networks. In our PD networks, we incorporate TC neurons to evaluate the effectiveness of aDBS. Within the setup of our PD networks, the TC fidelity is compromised, with many instances of the TC neurons failing to respond in a one-to-one fashion to excitatory inputs [21,37]. We study how closed-loop aDBS is able to improve TC relay fidelity when properly applied in PD networks. This represents the first work in which the effectiveness of closed-loop aDBS is evaluated based on TC relay fidelity. These results support the idea that adaptive stimulation of STN utilizing an interspike time merits further consideration as a possible alternative to standard forms of DBS for PD.

## 2. Results

We will be analyzing the proposed adaptive protocols under two different computational parkinsonian network configurations. The first configuration will consist of a network receiving adaptive constant pulse DBS (acDBS) in which the STN neurons have a four-spike burst with an initial TC error index (defined in Section 4.1.3) of 0.39 and 0.37 for the first and second TC cells, respectively. The second network configuration will consist of a network receiving adaptive local field potential DBS (aLFPDBS) in which the STN neurons have a three-spike burst with an initial TC error index of 0.56 for both TC cells. Throughout both computational studies, we will be applying stimulation in a time period we call the treatment window, which will start at 2000 ms and last until 7000 ms. During this time period, the stimulation will be turned on and off according to our adaptive protocols. By using these two network configurations, we can determine how well the proposed adaptive protocols would work in mild to moderate as well as advanced computational parkinsonism. In evaluating the efficacy of acDBS and aLFPDBS, we will examine whether each protocol is able to de-burst the network and under which settings we will see the largest improvement in the error index in the TC relay. Additionally, we will study how the stimulation changes the total synaptic input of the internal segment of the globus pallidus (GPi) to the TC cells. All model simulations were conducted using XPPAUT, and all model simulation figures were made in MATLAB.

### 2.1. Adaptive Constant Pulse DBS

In Figure 1, we can see examples of the adaptive constant pulse stimulation profiles when using a stimulation strength of $a_{0}=-15$ and an interspike time threshold parameter intt=200 ms.

During the treatment window of 2000 to 7000 ms, the stimulation currents delivered to neurons 1–4 are different. While not shown, it is the case that all 16 STN neurons will have different times when the stimulation is turned on and off. As outlined in Section 4.2, we use the interspike time—the time between successive STN firings—to determine when the stimulation is turned on and off. This is determined by the threshold parameter intt. If the interspike time is larger than the threshold parameter intt, then the stimulation is turned off. Otherwise, the stimulation is turned on because we predict that a bursting dynamic is occurring, which we wish to disrupt. Once the stimulation is turned on, the bursting dynamic is broken, and the interspike interval becomes larger than the intt threshold and will shut off again. This mechanism allows for neurons coming from the same synchronous groups to receive an individual stimulation based on its own firing pattern. For instance, neurons 1 and 2 belong to the ${\mathrm{STN}}_{11}$ subgroup, while neurons 3 and 4 ${\mathrm{STN}}_{12}$. However, when studying the stimulation profiles shown in Figure 1, we can see that the times when the stimulation is on and off vary by individual neuron.

When the acDBS protocol is applied to the STN neurons during the treatment window of 2000 to 7000 ms, we observe that the stimulation is able to successfully break the synchronized bursting dynamics observed in our computational parkinsonian state (shown in figure in Section 4.1.3). We are only left with single spike activity at intermittent intervals. An fexample of the membrane potential of ${\mathrm{STN}}_{1}$, with its corresponding stimulation profile, is shown in Figure 2. In this instance, the adaptive protocol is working as intended. Examining Figure 2, during the treatment window from 2000 to 7000 ms, bursting activity is suppressed. Once the interspike time exceeds the threshold intt and stimulation is turned off, it will remain off until we see the next STN spike and then the stimulation will turn back on to disrupt any potential STN bursting. We apply this throughout the treatment window until we shut the stimulation off at 7000 ms and we see the parkinsonian state of the network resume.

The overall impact of applying the acDBS throughout the entire 16 STN neurons, along with the changes to the two TC neurons, is shown in Figure 3. Some intermittent single-spike synchronization only occurs in a subgroup of STN neurons. Under the current stimulation settings ($a_{0}=-16$, intt=250 ms), the error index for each TC cell is 0.14 and 0.23 for cells 1 and 2, respectively. By changing the synchronized bursting pattern of the STN neurons, we have substantially altered the total synaptic GPi (top traces in blue in Figure 3) input to the TC cells. In Figure 3 (Panels B and C), examining the membrane potentials of $TC_{1}$ and $TC_{2}$ (black traces) with the excitatory inputs (red traces), the number of failures to respond in a one-to-one fashion in the parkinsonian state (see figure in Section 4.1.3) is greatly improved in the treatment window.

The changes in STN firing behavior are further reflected in the histograms of $sg_{1}$ and $sg_{2}$ corresponding to overall GPi synaptic input to the TC neurons (defined in Section 4.1.2). The histograms in Figure 4 with the stimulation show that the distribution has more elements in bin 1 and almost none in bins 5 and 6. Without stimulation, as shown in figure in Section 4.1.2, the histograms have substantially more elements sorted into bins 5 and 6. In the parkinsonian configuration, the overall GPi synaptic input to the TC cell is very phasic and bursty. This can be seen before the treatment window (1000–2000 ms) and after the treatment window (7000–8000 ms) in Figure 4. During the stimulation period, the phasic and bursty firing pattern seen in the parkinsonian state is interrupted and replaced with a more random pattern. This population-level phenomenon is the result of downstream propagation of the debursted STN neurons. These results indicate that we have significantly altered the firing of the GPi, replacing it with a more randomized firing. The corresponding changes in total GPi synaptic input to the TC cells improve the TC relay responses.

Having successfully broken the network parkinsonian state, we next seek to determine how robust this procedure is. Specifically, we seek to identify an effective range of values for the parameters $a_{0}$ and intt (the stimulation strength and the interspike time, respectively), under which the network will respond to the stimulation and reduce the TC error index. We are interested in finding a regime of optimal TC performance with relatively weak stimulation amplitude.

As shown in Figure 5, we can see well-separated regions where the error-index is high (0.5 and above), low (0.2 and below), and similar to the parkinsonian state (between 0.3 and 0.4). In contrast with previous work conducted by Guo and Rubin [21], there is a large range of values of interspike threshold times and stimulation strengths that desynchronize and deburst the STN neurons and therefore improve the TC performance. In general, the favorable region of stimulation parameters can be observed in the lower right-hand corner of Figure 5, with stimulation strengths ranging from −11 to −16 and interspike threshold parameters of 250 to 400 ms. The $\left(intt,a_{0}\right)$ parameter pairs in this window behave similarly to our results discussed above.

When studying Figure 5, we observe that pairing strong stimulation strength with lower interspike interval parameters produces non-optimal results. In these instances, our computational modeling shows that the stimulation to the neurons is occurring in a fashion that induces a different STN bursting pattern, which results in poor TC performance. For example, the $\left(intt,a_{0}\right)$ pair (75, −24) has an error index of 0.69 for $TC_{1}$. In Figure 6, we compare the firing of ${\mathrm{STN}}_{1}$ with its corresponding stimulation profile. Here we see that the firing in the STN neuron now comes in regular succession. In fact, the new firing pattern is directly induced from the stimulation provided to the STN cell, largely due to the strong stimulation strength and the short interspike intervals.

To better understand the impact that the stimulation-induced firing has on TC performance, we need to study how the induced firing impacts the total GPi synaptic input into the TC neurons. Using the $\left(intt,a_{0}\right)$ pair (75, −24), we compute $sg_{1}$ and $sg_{2}$ and construct the corresponding histograms. As seen in Figure 7, the total GPi synaptic input to $TC_{1}$ and $TC_{2}$ is indeed altered from the baseline parkinsonian network with no stimulation. However, unlike the $\left(intt,a_{0}\right)$ pair (250, −16), during the treatment window, we observe that the total synaptic GPi input is still clustered and synchronized. The histograms do not have a pronounced shift in bin 6 to bin 1. While the distribution is shifted in comparison to figure in Section 4.1.2, there are more entries in the middle bins 2–5.

This demonstrates that the TC performance is directly attributed to our choice of acDBS parameters. Further, when studying Figure 5, it also suggests that there is a non-linear relationship to how changes in the $\left(intt,a_{0}\right)$ space impact TC performance. In general, we can conclude that there are many combinations of stimulation strengths and interspike threshold parameters that are able to produce improved TC response in acDBS. We see that when the interspike threshold parameter is larger, we can use lower stimulation strengths, which is preferred. Conversely, pairing strong stimulation strengths with a short interspike threshold parameter produces a new firing dynamic bad for TC performance. Thus, there are numerous pairings of parameters, which may induce a theoretical therapeutic benefit.

### 2.2. Adaptive Multi-Site LFP Stimulation

Unlike the acDBS method discussed above, the adaptive LFP stimulation described by Equation (Equation 19) represents a more sophisticated closed-loop DBS. While acDBS is able to stimulate only when necessary, it is unable to adapt the stimulation strength to the amount of abnormal neuronal synchronization. We overcome this limitation by incorporating the recorded LFP signal of the STN neurons. Previous research has assessed the performance of STN stimulation based on recorded LFP signals in terms of its desynchronizing effects on model neurons [13,21,22,31]. This previous work shows that, while there is no clear evidence on how the LFP is related to synaptic and ionic currents of a single neuron, such stimulation is able to greatly reduce phase synchronization [13,22,30,31].

In Figure 8, we can see examples of the aLFPDBS profiles when using a stimulation strength of $a_{0}=6$ and an interspike time threshold parameter intt=300 ms. As with the acDBS results described previously, the stimulation current delivered to neurons 1, 4, 6, and 11 is different during the treatment window of 2000 to 7000 ms. While not shown, it is still the case that all 16 STN neurons will have different times when the stimulation is turned on and off. These differences persist when the neurons are coming from the same stimulation site (Neurons 1 and 6 of the ${\mathrm{STN}}_{11}$ block) or from a different stimulation site but are part of the same synchronous group (Neuron 4 from the ${\mathrm{STN}}_{12}$ block and Neuron 11 from the ${\mathrm{STN}}_{21}$ block). In contrast with acDBS, the amount of current delivered during the stimulation on period is now modulated by the filtered LFP on a delay as the signal is shuffled through the four stimulation sites (see figure in Section 4.3 and Equation (Equation 19)).

When the aLFPDBS protocol is applied to the STN neurons during the treatment window of 2000 to 7000 ms, we again observe that the stimulation is able to successfully break the synchronized bursting dynamics described in Section 4.1.3, and we are left with intermittent single spike activity. An example of the membrane potential of ${\mathrm{STN}}_{1}$ with its corresponding stimulation profile ($a_{0}$ = 10, intt = 325 ms) is shown in Figure 9. During the treatment window, the amount of stimulation delivered is modulated by the stimulation strength parameter $a_{0}$ and by the value of the LFP shown in Figure 10. Throughout the 2000–7000 ms period, the stimulation is turned off when the length between two successive spikes is larger than the interspike time threshold parameter intt. This allows for single spikes to occur before switching the stimulation on. When the length between two successive spikes is less than the interspike time threshold intt, the stimulation is on to prevent any potential bursting activity.

The overall impact of applying the aLFPDBS throughout the entire 16 STN neurons, along with the changes to the two TC neurons, is shown in Figure 11. We are able to successfully deburst the network, and there is intermittent single-spike synchronization present in some subgroups of STN neurons. Under the stimulation settings $a_{0}=6$, intt=300 ms, the error index for each TC cell is 0.13 and 0.2 for cells 1 and 2, respectively. This is a significant reduction from the network parkinsonian state, which had an error index of 0.56 for both model TC neurons.

The changes induced in the STN firing from the aLFPDBS are further reflected in the histograms of $sg_{1}$ and $sg_{2}$ corresponding to the overall GPi input to the TC neurons. Before the stimulation is turned on and the network is in the parkinsonian state, the GPi input to the TC cells is phasic and bursty. When aLFPDBS is applied, this phasic and bursty input is disrupted and replaced with a more random pattern consistent with what we observed when applying acDBS. Comparing the histograms in Figure 12 and figure in Section 4.1.2, we see that the distribution shifts from having more elements in bins 5 and 6 to more elements in bin 1, resulting in very few elements sorted into bins 5 and 6. This is attributed to the aLFPDBS significantly altering the STN firing pattern producing population-level neuronal changes. The debursting and desynchronization of the STN neurons propagate through the STN-GPe loop to the GPi, which, in turn, feeds into the two TC cells. By debursting and desynchronizing the STN neurons with aLFPDBS, we have greatly improved TC relay during the treatment window.

Having successfully broken the parkinsonian bursting dynamic, we will proceed to determine the robustness of aLFPDBS. We seek to identify an effective range of values for the parameters $a_{0}$ and intt under which the network will respond to the aLFPDBS and reduce the TC error index. We are interested in finding a regime of optimal TC performance with relatively weak stimulation amplitude.

When studying Figure 13, we see well-separated regions where the error-index is high (0.6 and above), low (0.2 and below), and similar to the network parkinsonian state (0.4 to 0.5). Consistent with our observations about acDBS, there is a large range of interspike threshold times and stimulation strengths that desynchronize and deburst the STN neurons and improves TC performance. The favorable region spans intt values from 250 to 400 ms and stimulation strengths $a_{0}$ ranging from 6 to 10. The $\left(intt,a_{0}\right)$ parameter pairs in this window behave similarly to the example of aLFPDBS shown in Figure 8, Figure 9, Figure 10, Figure 11 and Figure 12.

Under the aLFPDBS, pairing strong stimulation strengths with short interspike threshold parameters produces a train of periodic STN spiking throughout the treatment window, which results in pathological GPi outputs. For example, using the $\left(intt,a_{0}\right)$ pair of (100, 15) results in an error index of 0.81 and 0.72 for $TC_{1}$ and $TC_{2}$, respectively. As shown in Figure 14 and Figure 15, using a strong stimulation strength largely alters the amplitude of the LFP and disrupts the synchronized bursting of the STN neurons. However, the onset of more frequent and regular spiking that the stimulation induces in the STN neurons results in phasic and bursty GPi inputs to the TC cells. This pattern of firing in the STN and GPi corresponds to large error index values being observed. Another undesirable scenario occurs when $a_{0}$ is too small, for instance $a_{0}\leq 4$. In this scenario, we observe that aLFPDBS is not strong enough to break the bursting pattern. Furthermore, we lose the adaptive nature of the stimulation because the stimulation is on for the entirety of the treatment window.

### 2.3. DBS for Heterogeneous TC Cells

To conclude our analysis of this computational study, we investigated the robustness of the proposed adaptive protocols for restoring TC neuron relay responses. We aim to test whether the adaptive stimulations produce the same restoration results with variations of TC neurons. We completed this by generating 40 different model TC neurons with heterogeneity in the parameters $g_{L}$, $g_{Na}$, and $g_{T}$. Starting with the baseline values listed in Appendix A, we independently selected new parameters from normal distributions with standard deviations of 0.01, 0.05, and 0.08, as performed in [21,37]. All 40 different TC neurons receive identical GPi inhibition from the upstream basal ganglia loop. For each randomly generated set of parameters for TC neurons, we made a comparison between the parkinsonian network without stimulation and with stimulation. Since we have two parkinsonian networks under study, we paired the weaker parkinsonian configuration with acDBS. The stronger parkinsonian configuration was then compared with the aLFPDBS. It is critical to note that the parameter changes to the downstream TC neurons do not have any impact on the network parkinsonian condition. Based on the network configuration described in figure in Section 4.1, our model TC neurons do not connect back to the upstream STN-GPe loop or the GPi. The TC neurons only receive input from model GPi neurons. We would not expect underlying changes to our model TC neurons to impact the synchronized bursting of the STN neurons.

The results of the heterogeneous TC cell studies are shown in Figure 16. The first row displays the results of the acDBS study, while the second row displays the results of the aLFPDBS study. As seen in Panels A and B, Panel A shows 40 trials for $TC_{1}$ and Panel B shows 40 trials for $TC_{2}$. A majority of the trials for $TC_{1}$ and $TC_{2}$ show a moderate reduction in the TC error index from the parkinsonian network without stimulation to the network with acDBS. In Panels C (for $TC_{1}$) and D (for $TC_{2}$), the majority of the trials for the aLFPDBS show a significant reduction in the TC error index. We observe that the TC error index reduction in aLFPDBS is more significant than in acDBS. One possible explanation for the differences in TC error reduction is due to the nature of the stimulation delivered to the STN neurons as defined in Equations (Equation 16) and (Equation 19). While both of these methods are adaptive in that the stimulation is only turned on when necessary, the simplicity of acDBS limits its effectiveness. Comparing Figure 3A and Figure 11A, we observe that while acDBS is able to deburst the STN, there is a degree of isolated single spike STN synchronization. In comparison, when stimulating with aLFPDBS, the amount of isolated single-spike STN synchronization is further reduced.

## 3. Discussion

In this computational study, we investigated the effects of two different adaptive deep brain stimulation techniques—adaptive constant stimulation (acDBS) and adaptive local field potential stimulation (aLFPDBS)—and their effects on the thalamocortical relay. Here, we consider a network of synaptically-connected, conductance-based model neurons from the STN, GPe and GPi in the basal ganglia based on previous modeling work [21,37,38,39]. The model is parametrized to generate activity patterns featuring synchronized, rhythmic bursts fired by clusters of STN neurons, with different clusters bursting in alternation, which we take to represent a parkinsonian state. Outputs from the GPi are rhythmic (see figures in Section 4.1.2 and Section 4.1.3) and inhibit target model TC neurons that also receive excitatory input trains. We observe that the TC neurons are unable to respond reliably to these inputs, in agreement with earlier theory and simulations [21,37,39,40,41,42,43,44,45,46,47,48,49]. Using this framework, we considered two parametrizations of the parkinsonian state. The first network configuration consisted of a four-spike STN bursting pattern with a TC error index of 0.39 and 0.37 for the first and second TC cells, respectively. This configuration, which represents a mild to moderate state of computational parkinsonism, is used to investigate the acDBS mechanism. The second network configuration consisted of a three-spike STN bursting pattern with a TC error index of 0.56 for both model TC neurons. This configuration, representing an advanced state of computational parkinsonism, is used to investigate the aLFPDBS technique.

More recent computational studies on adaptive techniques have focused on stimulation within the STN that desynchronizes the rhythmic bursting dynamics found in the STN-GPe loop [13,24,25,26,27,28,29,31,32,33,34,35,36]. In this work, we demonstrate that the two proposed aDBS mechanisms are able to deburst the parkinsonian state and improve the TC error index while stimulation is applied. When these methods are properly tuned, the resulting model STN neurons exhibit single spike firing during the stimulation window. It is apparent that desynchronization is important to developing effective closed-loop DBS mechanisms; however, our study suggests that desynchronization of the STN neurons alone is not enough to improve thalamocortical relay in our PD networks. In our study, we found that debursting was critical to improving TC relay. Because of the elimination of the bursting dynamic, the combination of debursting and partial desynchronization of the single spikes of the STN neurons is sufficient to restore the TC relay. For both methods under study, our simulations demonstrate that the greatest improvements to the TC error index were achieved when we debursted and desynchronized the STN neurons.

In the present study, all model STN neurons in our PD networks received the same stimulation strength when testing both adaptive mechanisms. Further in silico studies of interest would involve stimulating synchronized subgroups of the STN neurons while leaving other parts of the network unstimulated. Similarly, conducting trials in which all synchronized STN neurons receive different levels of stimulation strength would also be of interest.

Another avenue of interest in aDBS methods is further investigating the detection mechanism for controlling when the stimulation is on and off. Currently, we are determining when stimulation is turned on and off based on an interspike time method. When the interspike time is above a preset threshold, the stimulation is turned off; otherwise, the stimulation is on. This method is an initial investigation for predicting when the STN neurons might exhibit a bursting dynamic we wish to interrupt. Finding novel ways to detect parkinsonian firing is an open problem with many possible avenues of research. Recently, work by Jung et al. in 2023 used whole-brain dynamic modeling and machine learning for the classification of PD [50]. Their study uses a Jensen-Rit model type of interacting excitatory and inhibitory neuronal populations. Their work demonstrates that personalized whole-brain models can serve as an additional source of information relevant to the diagnosis and possibly treatment of PD [50]. As aDBS becomes more widely utilized in personalized medicine and targeted therapies, there will be an increased need to identify beneficial biological feedback signals used in controlling the delivery of stimulation.

At the present moment, there is one commercially available brain stimulation system that provides closed-loop DBS [11]. While research on aDBS is still in its early stages, the preliminary findings suggest that aDBS is superior to the current standard open-loop HF stimulation being used [51,52,53]. Meta-analysis of the existing studies has proven to be challenging due to the heterogeneity of research methodologies and the small number of studies that have been conducted [52]. The issues surrounding the amount and quality of data available are not new and reflect a continuing challenge in studying STN DBS in Parkinson’s disease [54]. While these challenges will persist, experimenters and clinicians will increasingly need to rely on computational models to gain insights and correlations between neuronal activity and physical symptoms.

## 4. Methods and Materials

### 4.1. The Network Model

To develop a biologically faithful PD network model, we will adopt the same Hodgkin–Huxley model of basal ganglia thalamic network as in [21,37,38,39] with modifications to incorporate a variety of adaptive stimulation protocols of the STN neurons. Neurons in the basal ganglia and the thalamus communicate through various excitatory and inhibitory synaptic connections and receive certain external inputs. The basal ganglia circuit consists of GPe, GPi, STN neurons, and striatal input.

As depicted in Figure 17, both the GPi and GPe receive excitatory inputs from the STN. The GPe and GPi are subject to an inhibitory striatal input. There is synaptic coupling among inhibitory GPe neurons, and there is no coupling within the STN population and the GPi population. The TC cell is a relay station whose role is to respond under the GPi inhibition to incoming sensorimotor excitation via corticothalamic projections.

The network model consists of TC, STN, GPe, and GPi neurons. We will first describe the equations of STN, GPe and GPi neurons in the model network [37,38,39]. We then will present the TC neuron equations [37,39], which will be used to evaluate the DBS effectiveness. All specifics of the functions and parameter values used for each type of neuron in the model are given in Appendix A.

The STN voltage equation that we use takes the form
(1)$$\begin{array}{ccc} C_{m}v_{Sn}^{\prime } & = & -I_{L}-I_{Na}-I_{K}-I_{T}-I_{Ca}-I_{AHP}-I_{GPe\to STN}+I^{stim}, \end{array}$$
and was introduced in [38]. All the currents and corresponding kinetics are the same except that we make some parameter adjustments so that STN firing patterns are more similar to those reported in vivo [55,56,57]. $I_{GPe\to STN}$ is the inhibitory input current from GPe to STN. $I^{stim}$ is the external stimulation applied to STN.

The voltage of each model GPe neuron obeys the equation
(2)$$\begin{array}{ccc} C_{m}v_{Ge}^{\prime } & = & -I_{L}-I_{Na}-I_{K}-I_{T}-I_{Ca}-I_{AHP}-I_{GPe\to GPe}-I_{STN\to GPe}+I_{app}\left(t\right), \end{array}$$
where $I_{GPe\to GPe}$ is the inhibitory input from other GPe cells, $I_{STN\to GPe}$ is the excitatory input from STN cells, and $I_{app}\left(t\right)$ is a time-dependent external current that represents hyperpolarizing striatal input to all GPe cells.

The voltage equation for each model GPi neuron is similar to that for the GPe neurons, namely
(3)$$\begin{array}{ccc} C_{m}v_{Gi}^{\prime } & = & -I_{L}-I_{Na}-I_{K}-I_{T}-I_{Ca}-I_{AHP}-I_{STN\to GPi}+I_{GPe\to GPi}+I_{appi}\left(t\right), \end{array}$$
where $I_{STN\to GPi}$ represents the excitatory input from STN to GPi, $I_{GPe\to GPi}$ is the inhibitory input from GPe to GPi, and $I_{appi}\left(t\right)$ are time-dependent external inputs that represent hyperpolarizing currents from the striatum to all GPi cells. The time-dependent external currents $I_{app}\left(t\right)$ and $I_{appi}\left(t\right)$ are different from the constant inputs used in [21,37,38,39] and will take the form of a square-wave pulse given by
(4)$$I_{appi}\left(t\right)=q_{1}H\left(\sin\left(2\pi \left(t-t_{off,1}\right)t_{p,1}\right)\right))(1-H\left(\sin\left(2\pi \left(t-t_{off,1}\right)t_{p,1}\right)\right),$$
and
(5)$$I_{app}\left(t\right)=q_{2}H\left(\sin\left(2\pi \left(t-t_{off,2}\right)t_{p,2}\right)\right))(1-H\left(\sin\left(2\pi \left(t-t_{off,2}\right)t_{p,2}\right)\right).$$
Here, *H* is used to denote the Heaviside step function, such that H(x)=0 if x<0 and H(x)=1 if x>0. The parameters $q_{1}$ and $q_{2}$ in Equations (Equation 4) and (Equation 5) represent the amplitude of the square wave pulses and are consistent with the constant input values used in [21,37]. The parameters $t_{off,i}$ and $t_{p,i}$, i=1,2 are used to represent the period and duration of the square-wave pulses and will take on the values $t_{off,1}=12$ ms, $t_{p,1}=50$ ms, $t_{off,2}=6$ ms, and $t_{p,2}=40$ ms.

The model for each TC neuron takes the form
(6)$$\begin{array}{ccc} C_{m}v^{\prime } & = & -I_{L}-I_{Na}-I_{K}-I_{T}-I_{GPi\to TC}+I_{E}+c\left(t\right)\\ h^{\prime } & = & \left(h_{\infty }\left(v\right)-h\right)/\tau_{h}\left(v\right)\\ r^{\prime } & = & (r_{\infty }\left(v\right)-r)/\tau \left(v\right) \end{array}$$

In system (Equation 6), $I_{L}=g_{L}\left(v-v_{L}\right)$, $I_{Na}=g_{Na}m_{\infty }^{3}\left(v\right)h\left(v-v_{Na}\right)$, and $I_{K}=g_{K}{\left(0.75\left(1-h\right)\right)}^{4}\left(v-v_{K}\right)$ are leak, sodium, and potassium currents, respectively. We apply a standard reduction in our expression for the potassium current to decrease the dimensionality of the model by one variable [58]. The current $I_{T}=g_{T}p_{\infty }^{2}\left(v\right)r\left(v-v_{T}\right)$ is a low-threshold calcium current, where *r* is the inactivation and $p_{\infty }^{2}\left(v\right)$ is the activation. The membrane capacitance $C_{m}$ is normalized to 1 μF/cm^2^ in all the neural models included in the current work.

Additional terms in system (Equation 6) are inputs that the model TC neuron receives. One is the inhibitory input current from the GPi, $I_{GPi\to TC}$, such that
(7)$$\begin{array}{c} I_{GPi\to TC}=g_{GPi}s_{GPi}\left(v-V_{GPi}\right), \end{array}$$
where $g_{GPi}$ is the constant maximum conductance and $V_{GPi}$ is the synaptic reversal potential. $s_{GPi}$ satisfies the equation
(8)$$\begin{array}{ccc} s_{GPi}^{\prime } & = & \alpha_{GPi}\left(1-s_{GPi}\right)S_{\infty }\left(v\right)-\beta_{GPi}s_{GPi}, \end{array}$$
where $S_{\infty }\left(x\right)={\left(1+e^{-(x+57)/2}\right)}^{-1}$.

The other input to the model TC neuron, $I_{E}$, represents simulated excitatory sensorimotor signals to the TC neuron. We assume that these are sufficiently strong to induce a spike in the absence of inhibition and therefore may represent synchronized inputs from multiple presynaptic cells. We tune the parameters so that the TC cell yields spontaneous spikes at a rate of roughly 12 Hz in the absence of both inhibitory GPi and excitatory synaptic inputs. The parameter values chosen place the model TC neuron near the transition from silence to spontaneous oscillations. In the model, $I_{E}=g_{E}s\left(v-v_{E}\right),$ where $g_{E}=0.018$ mS/cm^2^, and *s* satisfies equation
(9)$$\begin{array}{ccc} s^{\prime } & = & \alpha (1-s)exc(t)-\beta s \end{array}$$
where α=0.8 ms^−1^ and β=0.25 ms^−1^. The function exc(t) controls the onset and offset of the excitation: exc(t)=1 during each excitatory input, whereas exc(t)=0 between excitatory inputs. The periodic exc(t) takes the following form:(10)$$\begin{array}{c} exc(t)=H(\sin(2\pi t/p))(1-H(\sin(2\pi (t+d)/p))), \end{array}$$
where the period p=50 ms and duration d=5 ms, and H(x) is the Heaviside step function, such that H(x)=0 if x<0 and H(x)=1 if x>0. Hence, exc(t)=1 from time 0 up to time *d*, from time *p* up to time p+d, from time 2p up to time 2p+d, and so on. A similar periodic function was used in previous work [37,39]. A baseline input frequency of 20 Hz is consistent with the high–pass filtering of corticothalamic inputs observed in vivo [59]; at this input rate, the model TC cells rarely fire spontaneous spikes between inputs.

In the following subsections, we focus on three aspects of the PD network model: the coupling structure in the STN–GPe loop, the averaged GPi synaptic input going into a TC relay neuron, and the TC relay error index.

#### 4.1.1. Architecture of Coupling between Individual Neurons

As shown previously in [38], the STN and GPe sub-network can generate both irregular asynchronous and synchronous activity [38,60,61]. Our model builds off of [37], where each STN, GPe, and GPi group includes 16 neurons. We incorporated two relay TC neurons into the parkinsonian network to evaluate the performance of DBS [37,39]. The network model mimics the pathological neuronal activity observed in the basal ganglia in parkinsonian conditions, such as increased firing rate, bursting patterns, and synchronization in STN and GPi neurons [40,41,42,43,44,45,46,47,48,49]. We consider this rhythmic clustered regime in STN and GPi as the parkinsonian state and refer to the network in this state as the parkinsonian network. In our simulation results that are presented throughout, we will discard the first 1000 ms to ensure that the network is in the parkinsonian state.

We designed the structure of the STN-GPe loop in the model following the work on clustered rhythms in [38] so that the STN cells will segregate into two rhythmically bursting clusters, with synchronized activity within each cluster. The detailed structure of connections between STN and GPe neurons, along with their connections to the remaining GPi and TC cells, is depicted in Figure 17. In the STN and GPe sub-network, there are both strong and weak synaptic connections built into the architecture of the network. This is reflected in Figure 17 by duplicating the 16 STN and GPe neurons to show the symmetry present in building the strong and weak connections necessary to create two synchronized groups of neurons. We use K*_ij_*, where K = GPe or STN, i=1,2, and j=1,2, to denote sub-population *j* within the *i*-th cluster of type K neurons. For example, the first sub-population of STN cluster one, STN_11_, sends excitation to the first sub-population of GPe cluster two, GPe_21_. The same sub-population of STN neurons are also weakly coupled with the other half of the same GPe cluster, GPe_22_. Each sub-population of STN neurons is connected with two GPe sub-populations in an analogous way. Each sub-population of GPe neurons inhibits one group of STN neurons, as is also illustrated in Figure 17. Within each GPe sub-population GPe*_ij_*, there are also local inhibitory connections.

The model also includes 16 GPi neurons, each receiving input from a single corresponding STN neuron. Thus, the rhythmic, bursty, synchronized outputs of each STN cluster induce rhythmic, bursty, synchronized activity in a corresponding group of GPi neurons. These GPi activity patterns mimic those seen experimentally in parkinsonian conditions. The network architecture is set up so that members of each such synchronized GPi group ($GPi_{1}$ or $GPi_{2}$) send synaptic inhibition to the same TC neuron, and hence each TC neuron receives a rhythmic inhibitory signal in the parkinsonian network (see Figure 17), which disrupts the fidelity of TC relay responses to excitatory inputs.

#### 4.1.2. Averaged GPi Synaptic Input to TC

In our network model, the synaptic input from the GPi to a TC neuron, $I_{GPi\to TC}$, comes from a subgroup of GPi neurons. As illustrated in Figure 17, the first GPi subgroup maps to the first TC cell and the second GPi subgroup maps to the second TC cell. Following [21,37], we will let $v_{TC_{j}}$ denote the membrane potential of the *j*-th TC cell. It follows that this input will take the form
(11)$$I_{GPi_{j}\to TC}=g_{GPi}\left(v_{TC_{j}}-v_{GPi}\right)\sum_{k\in \Omega_{j}}s_{GPi_{j}}^{k},\;j=1,2,$$
where each $\Omega_{j}$ is an index set for the neurons in the GPi group, $g_{GPi}$ is the maximal conductance, and $v_{GPi}$ is the synaptic reversal potential for inhibition from the GPi group. Each $s_{GPi_{j}}^{k}$ in Equation (Equation 11) satisfies the equation
(12)$$s_{GPi}^{\prime }=\alpha_{GPi}\left(1-s_{GPi}\right)S_{\infty }\left(\tilde{v}\right)-\beta_{GPi}s_{GPi},$$
where $S_{\infty }\left(x\right)={\left(1+e^{-(x+57)2}\right)}^{-1}$ and $\tilde{v}$ represents the membrane potential of the *k*-th GPi neuron from subgroup $GPi_{j}$.

Based on the structure of Equation (Equation 12), we can see that each $s_{GPi_{j}}^{k}$ is between 0 and 1. We define the quantities $sg_{1}$ and $sg_{2}$ by
(13)$$sg_{1}=\sum_{k\in \Omega_{1}}s_{GPi_{1}}^{k},$$
and
(14)$$sg_{2}=\sum_{k\in \Omega_{2}}s_{GPi_{2}}^{k}.$$
Since each $s_{GPi_{j}}^{k}$ is between 0 and 1, it follows that $sg_{1}$ and $sg_{2}$ are each between 0 and 8. In our computational study, we use the variability of the time-average of each $sg_{i}$ as an indicator of GPi rhythmic bursting activity. We do this by constructing histograms based on the frequency with which each $sg_{i}$ time series, averaged over 25 ms time windows, takes different values in bins that cover the range of [0, 8]. In analyzing both the parkinsonian state and the adaptive stimulation protocols, we will construct the histograms over the time window during which stimulation is applied from 2000 to 7000 ms. Specifically, we display six bins centered at 1 through 6, respectively, and each represents a subinterval of 1 ms/cm^2^, except that all values less than 1.5 are placed in the 1 bin and all values greater than 5.5 are placed into the 6 bin. In the parkinsonian network without external stimulation, the average $sg_{i}$ values fall into the 1 and 6 bins, as seen in Figure 18.

This result occurs because the GPi firing is both rythmic and bursty (see the top traces in Figure 19). Studying the top traces in blue in Figure 19 in panels B and C, we see that the GPi synaptic output is high during each bursting episode and low in between bursting events. A few values will be sorted into the middle bins 2 through 5 in Figure 18 due to the transition between bursting and quiescent phases. We will show that very different results emerge when our adaptive stimulation protocols are applied to the STN neurons (Figure 4, Figure 7 and Figure 12).

#### 4.1.3. TC Relay Responses and Error Index

Based on the network architecture described above, the synaptic input from the GPi to the target TC cells is both rhythmic and bursty (see Figure 19 Panels B and C, top curve). While there are instances where the TC neuron fires a single spike in accordance with the excitatory input that it receives (see Figure 19 Panels B and C, middle curve), other excitatory inputs to the TC neuron either result in no spiking activity or firing multiple spikes in response to a single excitatory input. This failure in one-to-one response between excitatory input and TC response is how we will characterize and define TC relay fidelity.

In our computational study, we quantify the relay performance of each TC neuron using a simple error index, which is computed by dividing the total number of errors—instances in which the TC cell either does not fire or fires multiple spikes in response to a single excitatory input—by the total number of excitatory inputs. That is, we define
(15)$$\mathrm{error}\;\mathrm{index}=\frac{b+m}{n},$$
where *n* is the total number of excitatory inputs. In Equation (Equation 15), *b* will represent the number of excitatory inputs in which the TC neuron gives a bad response consisting of more than one spike. Typically this will be in the form of a bursting episode but will also include a single-spike response followed after a delay, but before the next input, by additional spikes [37]. The number *m* in Equation (Equation 14) will represent the number of excitatory inputs that are “missed” by the TC neuron. That is, it is the number of excitatory inputs to the TC neuron that results in no corresponding spiking activity during a detection window [37]. Consistent with [37,39], the detection window used in our study extends from the beginning of each excitatory input to 18 ms after each input. This error index was first introduced in [39] and was used previously with the same error detection algorithm to quantify how different patterns of inhibitory GPi signals obtained from experimental recordings of normal and parkinsonian monkeys, with and without DBS [62], affect the TC relay response [37].

Adaptive deep brain stimulation (aDBS) is a closed-loop and demand-controlled method where DBS is turned on and off according to a feedback signal. In this approach, stimulation is administered only when necessary and to an extent dependent on the measured neuronal activity or symptoms [13]. Examples of stimulation trigger events or biophysical markers include action potentials from the targeted region of the brain and the amplitude of the beta-band local field potential (LFP) of the subthalamic nucleus (STN) measured via implanted electrodes [13]. In this study, we used the interspike time—the time between successive spikes—of the STN neurons to monitor the amount of ongoing abnormal neuronal activity to determine when stimulation will be applied to the network. We chose to use the interspike time between successive STN spikes as our biological signal due to several considerations. First, in our biophysical model, it is not difficult to detect when the action potential occurs. Second, this measurement is coming from the targeted region for stimulation and thus will form a closed feedback loop in the network. This feedback loop will help modulate the stimulation in real-time. Additionally, the use of the interspike time of successive STN neurons as a biological signal to control the adaptive delivery of stimulation has not been studied yet. Using this as our detection method, we aim to not only desynchronize the bursting dynamics of the parkinsonian network but to provide stimulation in such a way that the bursting dynamic of the STN neurons is eliminated completely, and we seek to improve the TC relay performance. The stimulation patterns that we will be testing under this adaptive scheme are constant pulse DBS (acDBS) and local field potential DBS (aLFPDBS) with coordinated reset shuffling.

### 4.2. Adaptive Constant Pulse DBS

The acDBS is given by the formula
(16)$$I_{k}^{\mathrm{stim}}=\sum_{i=1}^{N_{k}}a_{0}H\left(t-t_{i,1}\right)\left(1-H\left(t-t_{i,2}\right)\right),$$
where $a_{0}$ denotes the amplitude of the constant pulses, and *H* is the Heaviside function, where the times $t_{i,1}$ and $t_{i,2}$ are determined from the spiking mechanism of the STN neurons. These times are identified to track the interspike interval. For the adaptive protocol, if the interspike interval is larger than a preset threshold parameter, then the stimulation is turned off. After the stimulation has been turned off, stimulation will resume if the interspike interval is even smaller than the preset threshold. The corresponding turn-on time is $t_{i,1}$, and the off time is $t_{i,2}$. That is, when $t=t_{i,1}$, $H(t-t_{i,1})=1$ and $1-H(t,t_{i,2})=1$, and thus, the stimulation is on, and when $t=t_{i,2}$, $H(t-t_{i,1})=0$, and thus, the stimulation will be off.

### 4.3. Adaptive Multi-Site LFP Stimulation

Local field potentials (LFP) are transient electrical signals generated in nervous and other tissues by the aggregate electrical activity of the individual cells in that tissue (e.g., neurons). Since the LFP reflects the activity of many neurons in the vicinity of the recording electrode, it is therefore useful in studying local network dynamics.

Extracellular potentials are generated by transmembrane currents, and in the presently used volume conductor theory, the system is envisioned as a three-dimensional smooth extracellular continuum where the transmembrane currents are represented as volume currents [63]. In volume conductor theory, the fundamental formula for the contribution of extracellular potential ϕ(r,t) from the activity in an *N*-neuron model is given by
(17)$$\phi \left(r,t\right)=\frac{1}{4\pi \sigma }\sum_{n=1}^{N}\frac{I_{n}\left(t\right)}{|r-r_{n}|}.$$Here, $I_{n}\left(t\right)$ denotes the transmembrane current in compartment *n* positioned at $r_{n}$, and σ is the extracellular conductivity.

The measured raw LFP(t) is filtered online by applying a linear damped oscillator
(18)$$\ddot{x}+a\dot{x}+bx=\mu LFP\left(t\right).$$In the equation above, *b* approximates the frequency of the LFP oscillations and is expressed by b=2π/T where *T* is the mean period of the LFP. The parameters *a* and *b* are chosen in such a way that $a^{2}<4b$ to guarantee that Equation (Equation 18) represents a harmonic oscillator. The parameter μ controls the strength of the stimulation. We first choose the values of a=0.0025 and b=0.00136 so that the period of the harmonic oscillator is the same as the natural frequency of the bursts present in the STN clusters.

The aLFPDBS is given by the formula
(19)$$I_{k}^{stim}=\frac{\mu }{n}\sum_{i=1}^{N_{k}}\sum_{k=1}^{4}H\left(t-t_{i,1}\right)\left(1-H\left(t-t_{i,2}\right)\right)e^{-2dist(j,k)}x^{k}\left(t-\left(k-1\right)\tau \right),$$
where *n* is the number of STN cells, dist(j,k) is the distance between the *j*th neuron and the *k*th stimulation site, and $x^{k}\left(t-\left(k-1\right)\tau \right)$ is the time-delayed signal from Equation (Equation 18) that is delivered at the *k*th stimulation site, where τ is the delay and k=1,2,⋯4. Here, we calculate the distance in a two-dimensional Euclidian space. For instance, the distance between STN cell 2 and stimulation site 4 is given by $dist\left(2,4\right)=\sqrt{0.25^{2}+0.5^{2}}$. As before, the times $t_{i,1}$ and $t_{i,2}$ will be identified from the spiking mechanism of the STN neurons to track the interspike interval needed to control the delivery of the adaptive stimulation.

This formula is applied to the STN neurons through four stimulation sites, as illustrated in Figure 20. The 16 model STN neurons are represented by solid circles arranged in a four-by-four grid centered at the plus in the center. The first row on the square grid is $STN_{1},STN_{2},STN_{3}$, and $STN_{4}$ from left to right. $STN_{5}$–$STN_{8}$, $STN_{9}$–$STN_{12}$, and $STN_{13}$–$STN_{16}$ are in rows 2, 3, and 4, respectively. These neurons are spaced in such a way that the horizontal and vertical distance between any two neurons is 0.1. The four small boxes in Figure 20 are the stimulation sites, numbered 1, 2, 3, and 4, proceeding clockwise from the left. These sites are arranged to be at the center of a smaller two-by-two block formed by the four nearest STN cells.

## Figures and Tables

**Figure 1 ijms-24-05555-f001:**
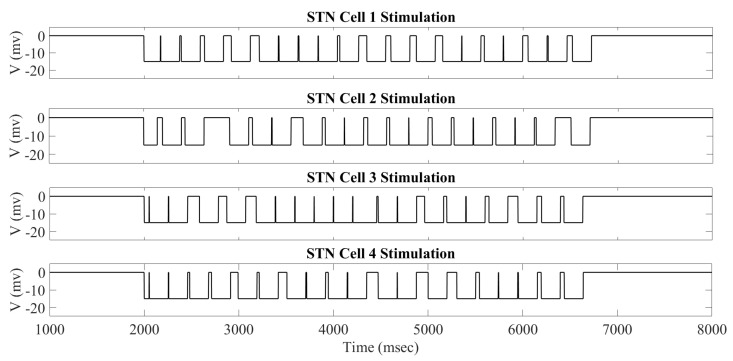
acDBS delivered to STN Cells 1–4 during the time window of 2000 to 7000 ms. Note that the top part of the curve represents the time when the stimulation is turned off, while the bottom part of the curve is when the stimulation is turned on.

**Figure 2 ijms-24-05555-f002:**
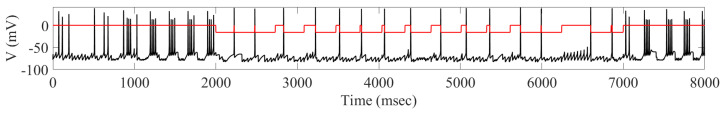
${\mathrm{STN}}_{1}$ membrane potential (black) with corresponding adaptive stimulation received (red) with parameters $a_{0}=-16$ and interspike time threshold parameter intt=250 ms. Note that the top part of the red curve represents the time when the stimulation is turned off, while the bottom part of the curve is when the stimulation is turned on.

**Figure 3 ijms-24-05555-f003:**
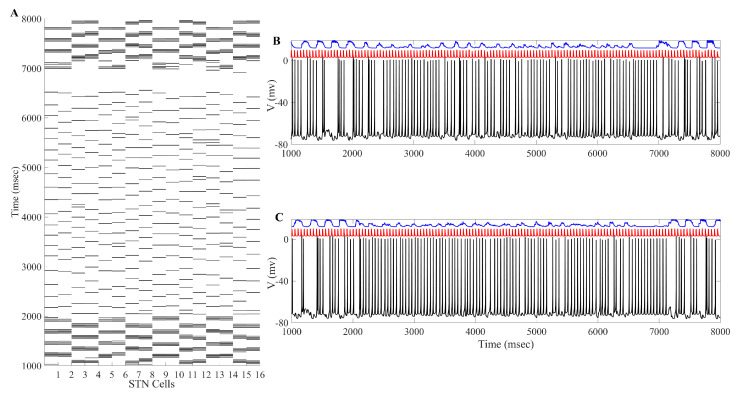
Example of acDBS. (**A**) Spike times of 16 STN neurons. acDBS is on from 2000 to 7000 ms with $a_{0}=-16$ and intt=250 ms. Before stimulation, there are two synchronized clusters, as shown in figure panel (a) in Section 4.1.3. During the stimulation period, the bursting dynamic is largely eliminated, and the synchronized clusters no longer exist. Once stimulation is stopped, the synchronized clusters reemerge. (**B**) Relay performance of ${\mathrm{TC}}_{1}$. (**C**) Relay performance of ${\mathrm{TC}}_{2}$. In both (**B**,**C**), the top trace in blue is the total GPi synaptic input, the middle trace in red is the excitatory signal, and the bottom trace in black is the TC voltage. TC relay performance noticeably improves during the stimulation window.

**Figure 4 ijms-24-05555-f004:**
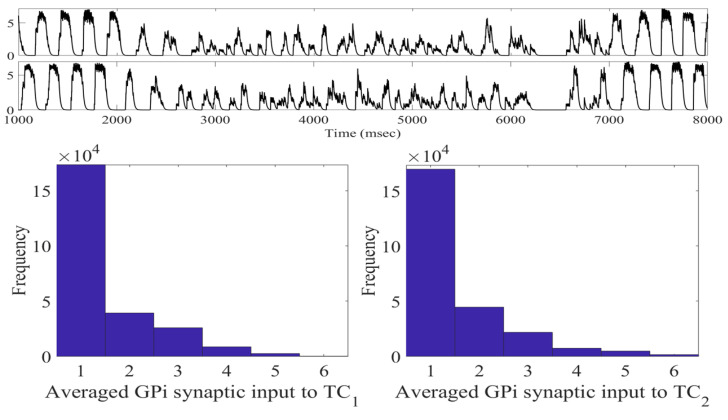
Total GPi synaptic inputs $sg_{1}$ and $sg_{2}$ (**top traces**) with corresponding histograms of $sg_{1}$ (**left**) and $sg_{2}$ (**right**) during the treatment window when acDBS was applied with $a_{0}=-16$ and intt=250.

**Figure 5 ijms-24-05555-f005:**
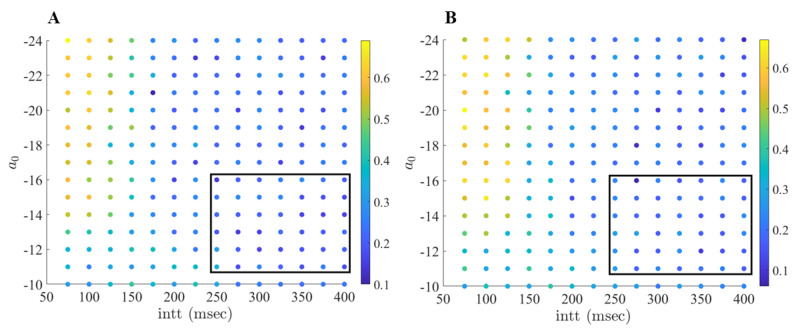
(**A**) Error index values (color coded) for the first TC cell, $TC_{1}$ over a range of interspike threshold parameter values intt (75–400 ms, with increment 25 ms) and $a_{0}$—stimulation strength (−24 to −10, with increments of 1). (**B**) Error index values for the second TC cell, $TC_{2}$ over a range of values intt (75 to 400 ms, increment 25 ms) and $a_{0}$ (−24 to −10, with increment 1). The favorable region of $\left(intt,a_{0}\right)$ pairs is shown in the black box.

**Figure 6 ijms-24-05555-f006:**
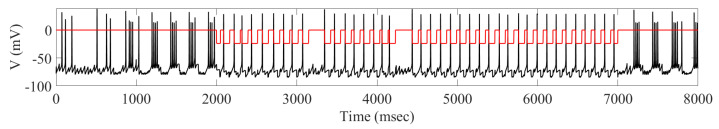
$STN_{1}$ membrane potential (black) with corresponding adaptive stimulation received (red) with parameters $a_{0}=-24$ and interspike time threshold parameter intt=75 ms. Note that the top part of the red curve represents the time when the stimulation is turned off, while the bottom part of the curve is when the stimulation is turned on.

**Figure 7 ijms-24-05555-f007:**
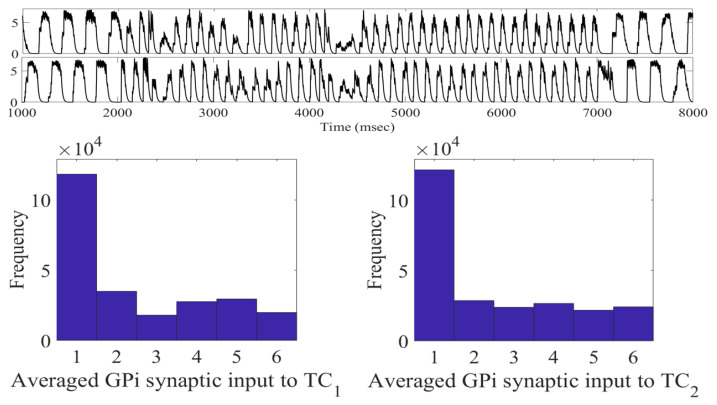
Total GPi synaptic inputs $sg_{1}$ and $sg_{2}$ (**top traces**) with corresponding histograms of $sg_{1}$ (**left**) and $sg_{2}$ (**right**) during the treatment window when acDBS was applied with $a_{0}=-24$ and intt=75.

**Figure 8 ijms-24-05555-f008:**
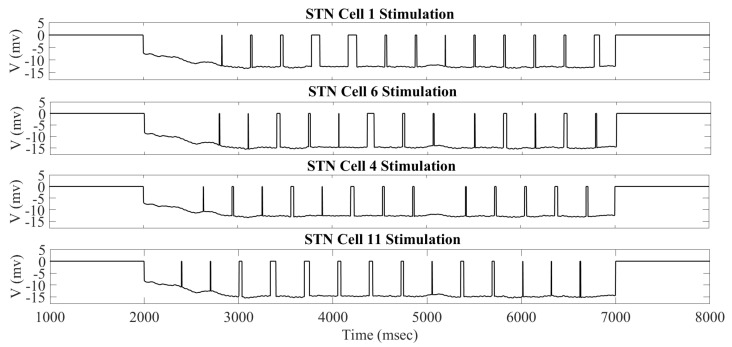
Adaptive local field potential stimulation delivered to STN Cells 1, 4, 6, and 11 arranged by the synchronous group during the time window of 2000 to 7000 ms. Note that the top part of the curve represents the time when the stimulation is turned off, while the bottom part of the curve is when the stimulation is turned on.

**Figure 9 ijms-24-05555-f009:**
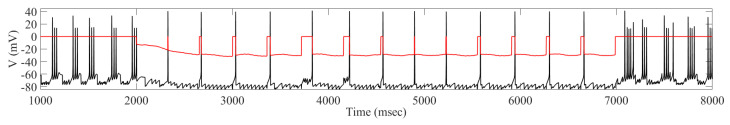
$STN_{1}$ membrane potential (black) with corresponding stimulation received (red) with parameters $a_{0}=10$ and interspike time threshold parameter intt=325 ms. Note that the top part of the red curve represents the time when the stimulation is turned off, while the bottom part of the curve is when the stimulation is turned on.

**Figure 10 ijms-24-05555-f010:**
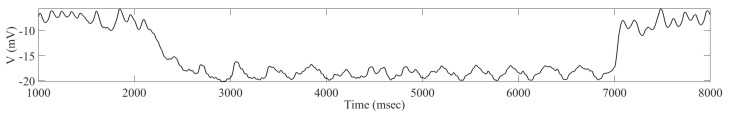
The total filtered local field potential of the 16 STN neurons from 1000 to 8000 ms with aLFPDBS parameters $a_{0}=10$ and intt=325.

**Figure 11 ijms-24-05555-f011:**
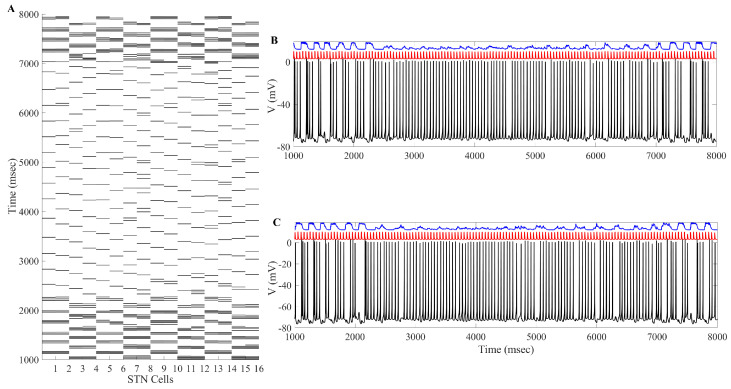
Example of aLFPDBS. (**A**) Spike times of 16 STN neurons. acDBS is on from 2000 to 7000 ms with $a_{0}=6$ and intt=300 ms. Before stimulation, there are two synchronized clusters, as shown in figure panel (a) in Section 4.1.3. During the stimulation period, the bursting dynamic is largely eliminated, and the synchronized clusters no longer exist. Once stimulation is stopped, the synchronized clusters reemerge. (**B**) Relay performance of ${\mathrm{TC}}_{1}$. (**C**) Relay performance of ${\mathrm{TC}}_{2}$. In both (**B**,**C**), the top trace in blue is the total GPi synaptic input, the middle trace in red is the excitatory signal, and the bottom trace in black is the TC voltage. TC relay performance noticeably improves during the stimulation window.

**Figure 12 ijms-24-05555-f012:**
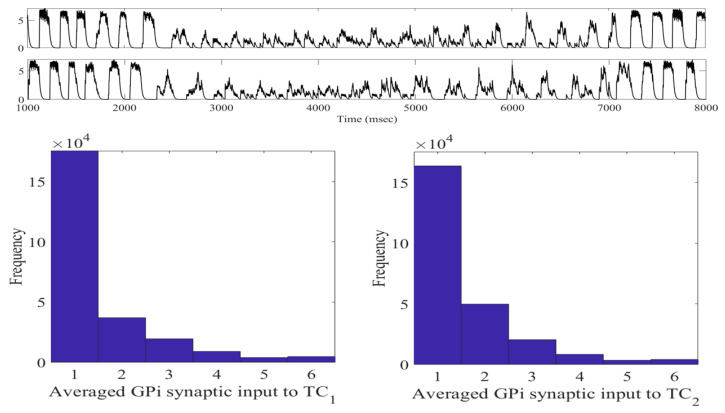
Total GPi synaptic inputs $sg_{1}$ and $sg_{2}$ (**top traces**) with corresponding histograms of $sg_{1}$ (**left**) and $sg_{2}$ (**right**) during the treatment window when aLFPDBS was applied with $a_{0}=6$ and intt=300.

**Figure 13 ijms-24-05555-f013:**
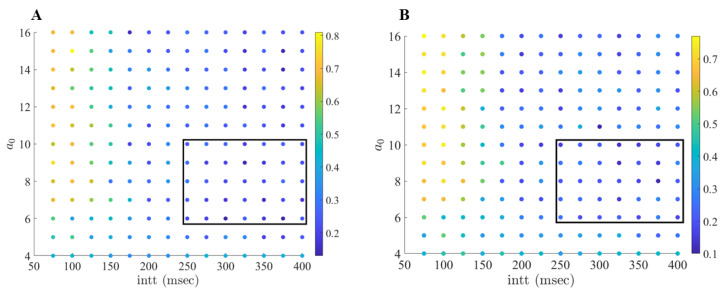
(**A**) Error index values (color coded) for the first TC cell, $TC_{1}$ over a range of interspike threshold parameter values intt (75 ms to 400 ms, with increment 25 ms) and $a_{0}$—stimulation strength (4 to 16, with increment 1). (**B**) Error index values for the second TC cell, $TC_{2}$ over a range of values intt (75 to 400 ms, increment 25 ms) and $a_{0}$ (4 to 16, with increment 1). The favorable region of $\left(intt,a_{0}\right)$ pairs is shown in the black box.

**Figure 14 ijms-24-05555-f014:**
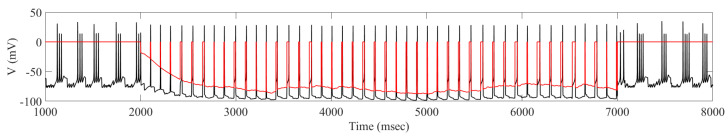
$STN_{1}$ membrane potential (black) with corresponding adaptive local field potential stimulation received (red) with parameters $a_{0}=15$ and intt=100. Note that the top part of the red curve represents the time when the stimulation is turned off, while the bottom part of the curve is when the stimulation is turned on.

**Figure 15 ijms-24-05555-f015:**
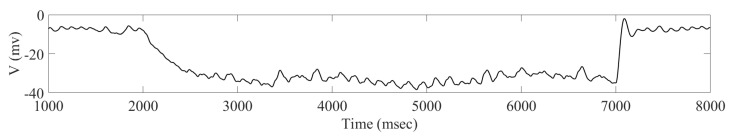
The total filtered local field potential of the 16 STN neurons from 1000 to 8000 ms with aLFPDBS parameters $a_{0}=15$ and intt=100.

**Figure 16 ijms-24-05555-f016:**
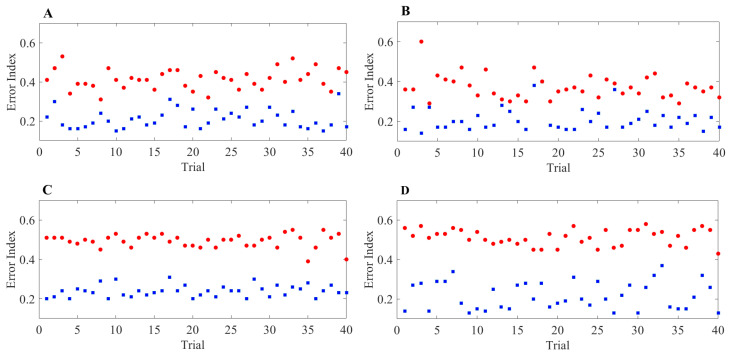
Error index values for 40 model TC neurons with heterogeneous TC parameter values. All baseline values of the TC parameter are given in Appendix A. (**A**) $TC_{1}$ heterogeneous error index values without stimulation (red) and with acDBS, Equation (Equation 16) (blue). (**B**) $TC_{2}$ heterogeneous error index values without stimulation (red) and with acDBS (blue). (**C**) $TC_{1}$ heterogeneous error index values without stimulation (red) and with acLFPDBS, Equation (Equation 19) (blue). (**D**) $TC_{2}$ heterogeneous error index values without stimulation (red) and with aLFPDBS (blue).

**Figure 17 ijms-24-05555-f017:**
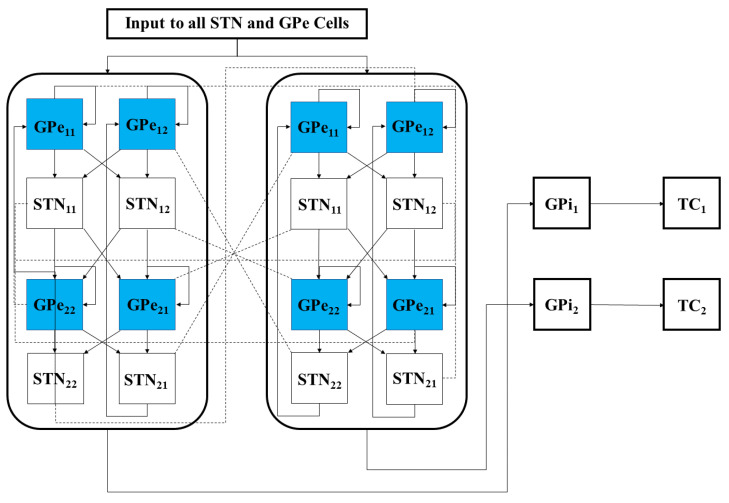
Full network model, including the interconnections between the STN and GPe sub-network neuronal populations. Each STN and GPe block represents 4 neurons, while the GPi 1 and GPi 2 blocks represent 8 neurons that connect to a single TC cell. The solid lines represent strong synaptic connections, while the dashed lines start from the STN block to the corresponding GPe block, representing weak synaptic connections.

**Figure 18 ijms-24-05555-f018:**
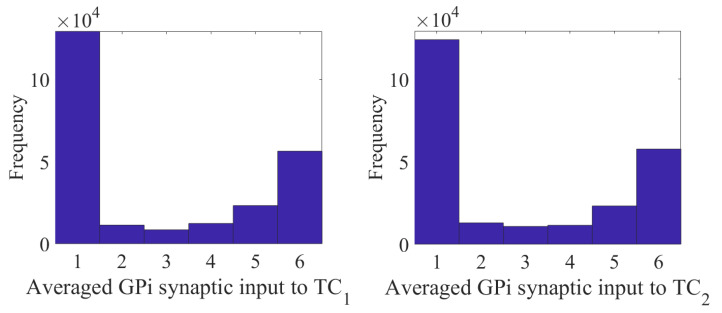
Histograms of $sg_{1}$ (**left**) and $sg_{2}$ (**right**) in the parkinsonian network, in ms/cm^2^. Both histograms include two dominant bins, centered at 1 and 6, due to the quiescent and bursting phases, respectively, of GPi activity.

**Figure 19 ijms-24-05555-f019:**
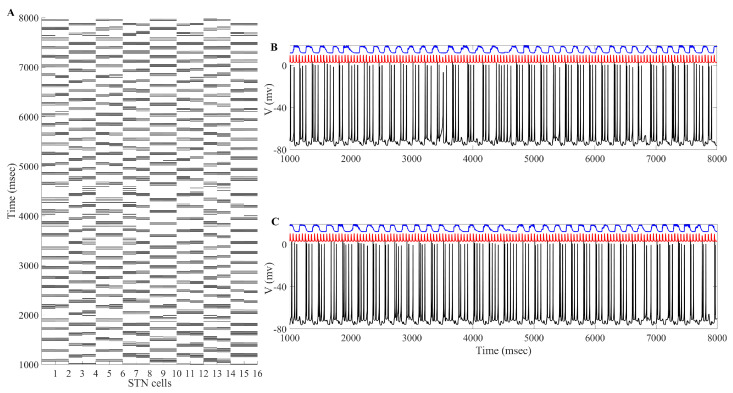
STN clusters in the parkinsonian network. Panel (**A**) shows that the 16 model STN neurons form two synchronized clusters. Panels (**B**,**C**) show the membrane potentials of the two TC neurons (bottom curve, black) responding to excitatory sensorimotor signals (middle curve, red), along with the total synaptic input the corresponding TC cell receives from eight GPi neurons (top curve, blue).

**Figure 20 ijms-24-05555-f020:**
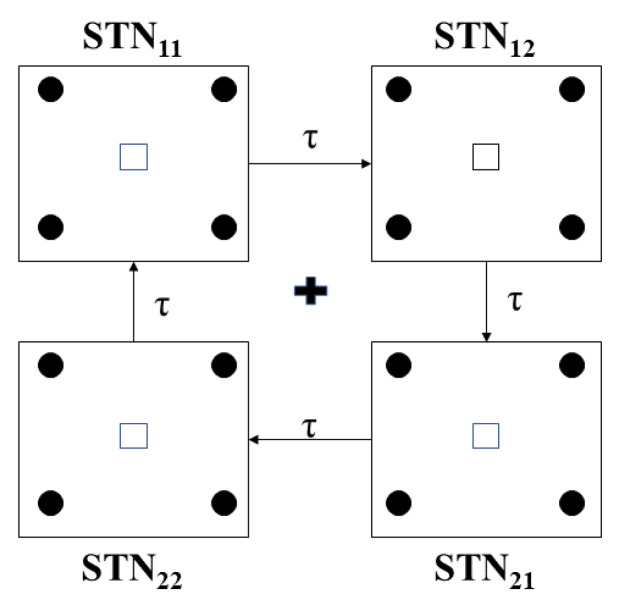
Sixteen STN cells (solid circles) on a square grid with the center (plus sign) where an electrode can measure the local field potential. The four square boxes are the stimulation sites where the signal will be shuffled through in a clockwise fashion. The signal will be on a time delay of τ through each site. In the acDBS protocol (Equation (Equation 16)), there is no delay and no shuffling of the stimulation.

## Data Availability

All the model simulations conducted in this study were completed using XPPAUT. The XPPAUT code is available for use by request to the corresponding author.

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
