# Peer review of "Adaptive Stimulations in a Biophysical Network Model of Parkinson’s Disease"

_ijms, 2023, doi:10.3390/ijms24065555_

Round 1
Reviewer 1 Report
In this exceptionally important paper, the Authors carried out a computational study to implement a novel technique of deep brain stimulation, where they have been able stimulate the subthalamic nucleus in an adaptive fashion using the inter-spike firing of the neurons as a biomarker to control stimulation. Their results show that such a protocol eliminates bursts in the synchronized bursting neuronal activity of the subthalamic nucleus which is hypothesized to cause failure of thalamocortical neurons to respond properly to excitatory cortical inputs. Further, they are able to significantly decrease the thalamocortical neurons relay errors, representing potential therapeutics for Parkinson’s disease. The paper shows extremely interesting results, is very original, clearly written and the results obtained by the Authors well support the conclusions they draw. It deserves publication as it is.
Reviewer 2 Report
Stojsavljevic et al., presented a network based simulated model of DBS-STN network to target the challenges and complications of high frequency stimulation.
Based on that my comments are:
1. Biological validity and clinical implications of a simulated model is always questionable and critical to interpret. Moreover this manuscript lacks its biological validity
2. Authors need to mention the challenges of High Frequency stimulations which can be addressed by this adaptive stimulation
3. Clinical implications of this adaptive stimulation is critical to address.
4. A pilot patients based study on respective adaptive stimulation is needed to validate this findings.
5. Clinical complications after adaptive stimulations and related ethical implications need to be addressed
Minor comments:
1. Introduction can be shorten and specific to the study
2. Some typo and grammatical editing are needed
Reviewer 3 Report
The present submission is a computational modeling study that designs an adaptive closed-loop deep brain stimulation for Parkinson’s disease. The paper is well prepared with many good visualization and details formulas. I provide some minor points for the authors to consider.
INTRODUCTION
・ As the present article is a computational modeling study (protocol), I would suggest the authors shorten the disease and pharmacological treatment background related to PD. The authors can add more information related to DBS treatment and its related disease pathology and treatment mechanisms.
・ Open-loop and closed-loop are important concepts, and their detailed definitions need to be included in the introduction.
・ The authors cited many references in one sentence. For example, “Additionally, battery and device shelf-life concerns increase with higher frequency and open-loop systems [2,9–16].” Please check if the citations are necessary.
・ Line 49: “A PD network” the term is confused. What is the network? Does it mean the disease-related loop?
METHODS
・ The rationale, for using inter-spike time, as a biomarker for triggering the stimulation, was not described.
・ The authors should be careful when using “biomarkers”. In this paper, it seems to me that the authors use a certain feature of the neural signal as the “trigger” in a closed-loop stimulation system. It may not be relevant to “biomarker”.
・ In the section on “Network Model”, it would be clearer to the reader, if the authors can add a schematic figure for the original PD network model first.
RESULTS:
・ There are more than 20 figures attached to the manuscript. The authors may consider reducing the number of images or pooling different plots in one graph.
Round 2
Reviewer 2 Report
Validation of computational model in disease biology is critical to interpret. A clinical collaborative study of this computational modelling can be helpful.